# The Role of Exercise Training in Delaying Kidney Function Decline in Non-Dialysis-Dependent Chronic Kidney Disease

**Mark Davies** [1,2,*], **Aamer Sandoo** [1] and **Jamie Macdonald** [1,*]

1   School of Human and Behavioural Sciences, Bangor University, Bangor LL57 2DG, UK; a.sandoo@bangor.ac.uk

2   Renal Department, Ysbyty Gwynedd Hospital, Bangor LL57 2PW, UK

*   Correspondence: mdavies@bangor.ac.uk (M.D.); j.h.macdonald@bangor.ac.uk (J.M.)

**Abstract:** Chronic Kidney Disease (CKD) is a progressive condition characterised by declining eGFR and associated, particularly in advanced stages, with increased morbidity and cardiovascular mortality. Current treatment options for delaying disease progression are limited to a small number of pharmacological agents. Considering that rates of kidney function decline are greater in patients with lower levels of habitual physical activity, there is interest in the potential benefits of structured exercise training in delaying CKD progression. This discursive review summarises the current state-of-play in this field of research by critically analysing the published systematic reviews of randomised controlled trials of structured exercise training in the non-dialysis CKD population. Several issues are highlighted that hamper definite conclusions as to the therapeutic effectiveness of exercise training for this purpose. However, following an overview of the pathophysiology and risk factors for kidney function decline, several potential mechanisms explaining how exercise training may benefit CKD progression are offered. Finally, suggestions for future research in this area are made. The review concludes that there is a need for further research on the effectiveness of exercise before it can be recommended as part of routine care for the purpose of delaying CKD progression. Exercise can be recommended, however, to individual patients because of a potential benefit to kidney function, and definite benefits to other outcomes such as quality of life, with no apparent evidence of harm.

**Keywords:** aerobic; Chronic Kidney Disease; disease progression; eGFR; exercise; kidney function; non-dialysis dependent; resistance; training

## 1. Introduction

Chronic Kidney Disease (CKD) is a progressive condition characterised by variable, but usually inevitable, annual rates of estimated glomerular filtration rate (eGFR) decline [1]. Even before progressing to end-stage kidney disease (ESKD), CKD is associated with increased mortality, particularly cardiovascular death [2]. Furthermore, CKD is an independent risk factor for cardiovascular disease with an increased risk of death, cardiovascular events and hospitalisation as eGFR falls [3]. Current treatment options for delaying the progression of CKD are limited to a small handful of therapeutic options, namely renin-angiotensin-aldosterone (RAS) system inhibitors, sodium bicarbonate supplementation and sodium-glucose transporter 2 (SGLT-2) inhibitors [4,5]. In the hunt for additional strategies, exercise training has been considered a possible therapy for delaying disease progression.

Despite evidence of exercise-induced acute kidney injury (AKI) in endurance athletes and increased proteinuria post-exercise, concerns that exercise may have a detrimental effect on kidney function have been allayed by a wealth of studies showing exercise training to be safe [6–9]. Indeed, structured exercise training has been shown to have multiple beneficial effects in the non-dialysis-dependent CKD (ND-CKD) population, including improved aerobic capacity [10–13], physical function [9,14,15], muscle strength [14–16]

and health-related quality of life (HRQoL) [9,17]. Increased habitual physical activity is associated with better survival in both ESKD [18] and ND-CKD patients [19,20] (though there is a possible influence of selection bias and other factors not adjusted for in these observational studies).

With regards to disease progression, there is observational data that links low physical activity with faster CKD progression. People with CKD are generally less active than the general population [21], with physical activity levels reduced further in advanced disease stages [22–25]. More than that, lower physical activity levels appear to be predictive of kidney function decline [26–28].

Therefore, understanding whether structured exercise can delay CKD progression will be of benefit to patients and practitioners. This narrative review briefly outlines the pathophysiology of kidney disease progression and highlights several key risk factors for kidney disease progression. The current evidence for the effect of exercise training on eGFR is reviewed before considering plausible ways in which exercise may help delay CKD progression. Future considerations for scientific studies are also suggested.

## 2. The Pathophysiology of CKD Progression

Whilst a thorough description of the pathophysiology of CKD progression is beyond the scope of this review and is available elsewhere (e.g., [29–32]), an outline of the key processes and mediators will aid the reader in understanding the potential beneficial effects of exercise training.

It is widely acknowledged that, regardless of the original aetiology of kidney disease, there are common subsequent pathways that ultimately lead to kidney fibrosis and loss of function [32–35]. Most kidney diseases begin in the glomeruli [32,36], with damage leading to several downstream effects. Firstly, there is a propagation of injury within the glomeruli, including to the podocytes, a key cell in the filtration barrier [37]. Podocyte injury results in increased passage of protein into the tubules, which in turn, results in increased protein uptake and processing by tubular cells. This stimulates proximal tubular cells into assuming a pro-inflammatory phenotype with altered transcription factor activity, including increased NF-κB with a resultant production of a number of pro-inflammatory and pro-fibrotic mediators, including MCP-1 and TGF-β1 [36,38]. Damage to glomeruli also leads to hyperfiltration of intact glomeruli in an attempt to compensate for the loss in number of functioning glomeruli [33,39]. This adaptive mechanism has negative consequences, including damage to glomerular cells and increased proteinuria. Furthermore, it compromises tubular blood supply both by glomerulosclerosis, with a consequent collapse of capillaries, and by haemodynamic changes (increased arteriolar resistance leading to reduced peritubular capillary blood flow) [32,35]. Hyperfiltration also leads to increased tubular activity, including the metabolically costly process of sodium reabsorption. The increased oxygen demand may lead to oxidative stress but also to relative cortical hypoxia [32,35]. Tissue hypoxia also results from capillary rarefaction: inflammatory processes in the tubulo-interstitium that result in a loss of endothelial integrity and cell death. The importance of a healthy endothelium has been highlighted by a number of experimental models which remove the key endothelial cell product nitric oxide (NO), produced by the enzyme endothelial nitric oxide synthase (eNOS): these include eNOS knockout diabetic mice, which show more rapid eGFR decline than wildtype [40]. The reduced blood supply caused by capillary rarefaction is exacerbated by the build-up of the interstitial fibrotic matrix (due to the inflammatory processes mentioned above), which increases the distance between surviving capillaries and the tubules they are supplying, thus worsening parenchymal hypoxia [41,42]. These various pathways of tubular injury lead to inflammatory processes that perpetuate further injury in all nephron compartments resulting in interstitial inflammation and fibrosis and loss of kidney function [36,43].

Driving these pathological processes is an imbalance of various mediators. Those briefly outlined here are amongst the most well-recognised and are relevant for this discussion about the potential effects of exercise. The importance of preserving tubular and

glomerular endothelium has been highlighted. Loss of vascular endothelial growth factor (VEGF), produced by both podocytes and tubular epithelial cells and important for maintaining endothelial integrity, has been implicated in furthering the progression of kidney damage [41,44], though it should also be noted that aberrant VEGF activity may be fibrogenic [45]. Angiotensin II (AngII), originally known primarily as a haemodynamic actor, has pleiotropic effects and is a key mediating molecule of many pathological pathways of kidney disease progression, including increasing proteinuria through direct effects on the filtration barrier, stimulating key inflammatory pathways and inducing oxidative stress [46]. Oxidative stress, an imbalance in pro-oxidant and anti-oxidant molecules, has been shown in experimental models to contribute to CKD progression both by direct cellular damage and by stimulating pro-inflammatory processes [47,48]. Prominent imbalances in inflammatory pathways that promote kidney damage include increased expression of pro-inflammatory transcription factors, particularly NF-κB [31,49], and broadly reduced anti-inflammatory Nrf2 (nuclear factor erythroid 2–related factor 2) [31]; chemokines, particularly MCP-1 (monocyte chemoattractant protein 1) and RANTES (Regulated upon Activation, Normal T Cell Expressed and Presumably Secreted) [29,30]; and cytokines, including IL-1β, Il-6, TNF-α [50,51]. Finally, sympathetic nervous system (SNS) overactivity, which is common in patients with CKD [52], has been implicated as a potential contributor not only of CKD-related hypertension, but also of disease progression [53,54], possibly by damaging peritubular capillaries leading to ischaemia [55].

## 3. Risk Factors for CKD Progression

There are several well-established risk factors for kidney disease progression, regardless of the original aetiology. General population studies [56–58] and observational studies in patients with established CKD [59–61] have demonstrated a marked worsening of renal outcomes in patients with hypertension; the risk increases as blood pressure increases. Additionally, there is evidence of benefit, i.e., slower eGFR decline, with improved blood pressure control in interventional studies [62,63]. In addition to hypertension per se, recent studies also identify that raised short-term systolic blood pressure variability (which is prevalent in patients with CKD [64] and partly due to increased sympathetic nervous activity [65]) is predictive of disease progression (rapid eGFR decline [66] or ESKD incidence [67]) in patients with established CKD.

The usual processes of renal blood flow autoregulation protect the glomerulus from systemic hypertension. Renal autoregulation, however, can be overwhelmed by sustained or severe hypertension [68,69]. Even in situations where autoregulation is still effective, hypertension can result in progressive renal damage in cases where there is underlying renal injury of another cause [69]. In both these situations, barotrauma, i.e., damage caused by pressure, results from persistent or severe hypertension. Examples of resultant damage include upregulation of fibrotic repair mechanisms due to capillary and mesangial cell stretching, worsening proteinuria (with its downstream effects) [70] and shedding or effacement of podocytes [71]. In addition to direct pressure effects, there is evidence that hypertension upregulates or exacerbates other pathogenic pathways, including endothelial dysfunction [72] and oxidative stress [73].

Similarly, obesity or raised body mass index (BMI) are strongly associated with adverse renal outcomes in general population studies and CKD cohorts [74–76]. Weight loss interventions, including surgical, pharmacological and lifestyle interventions, are able to prevent renal function decline and even improve eGFR in patients with CKD [77–79].

There are two broad mechanisms whereby obesity contributes to CKD progression. Adipocytes (fat cells) are not just storage cells, but produce various hormones, cytokines and other pro-inflammatory molecules, including angiotensinogen (the precursor to Ang II), TNF-α and IL-6 [80]. Altered levels of these molecules occur in obesity and may be contributory to renal damage [80]. Secondly, both animal and human studies have demonstrated that BMI in overweight or obese categories leads to increased renal plasma

flow and glomerular hyperfiltration [81]. As discussed above, hyperfiltration is linked to progressive glomerular damage and proteinuria.

Diabetes is the most common cause of CKD worldwide [82]; the presence of diabetes increases risk of ESKD and risk of incident moderate CKD in population studies [57,74,83], as well as increased risk of eGFR decline in observational studies in people with established CKD [84]. Glycaemic control has been established as a risk factor for diabetic kidney disease incidence and progression in randomised trials of strict control with various anti-glycaemic agents in both Type 1 [85] and Type 2 [86] diabetes mellitus.

The potential pathological effects of hyperglycaemia on the kidney are numerous and involve multiple cells of the nephron, particularly endothelial and mesangial cells. Hyperglycaemia stimulates various metabolic pathways, which are damaging when over-whelmed or inappropriately activated. These include stimulating TGF-β1 production, both directly and by protein kinase-C pathway activation [87], and aberrant mitochondrial activity with resultant oxidative stress production [88]. Advanced glycation end-products (AGEs), the products of non-enzymatic reactions between glucose and proteins or lipids, also form readily in hyperglycaemic states and can independently stimulate similar patho-logical pathways [87]. Hyperglycaemia also impairs renal autoregulation, and so augments the harmful effects of hypertension in the glomerulus [89].

Even in non-diabetic patients, insulin resistance is common in CKD [90]. Insulin resistance is a consequence of visceral obesity [91] and is highly related to metabolic syndrome [92], but there are additional factors in CKD—namely inflammatory and oxida-tive stress pathways, as well as metabolic acidosis—which promotes insulin resistance [90]. Both metabolic syndrome and insulin resistance have been implicated as risk factors for CKD development and progression [93,94].

Insulin resistance leads to hyperinsulinaemia. Raised insulin concentration, acting via the insulin-like growth factor-1 (IGF-1) receptor, stimulates vascular smooth muscle cell proliferation, mesangial cell growth and reduced metalloproteinase activity; the result is an alteration of glomerular structure and renal fibrosis [90]. Insulin can also impair renal autoregulation by a sodium-retaining action in the renal tubules, potentially leading to increased glomerular pressure [90]. It is also likely that insulin resistance reduces NO availability, thus having detrimental effects on renal haemodynamics [95].

Understanding the importance of the pathophysiology and risk factors for CKD progression is important for establishing plausible therapeutic targets and interventions. As will be reviewed later, the risk factors discussed here are modifiable by exercise intervention. Next, we will review the evidence concerning exercise training on CKD progression itself.

## 4. The Current State of the Research on Exercise Training to Delay CKD Progression

There has been, for some time, interest in the potential beneficial effects of exercise training on CKD progression per se. The earliest randomised trial in humans found no benefit [96]. Some other early studies in this field, however, showed reduced rates of decline, and even improvement, after aquatic [97], aerobic [98] and resistance [99] exercise interventions.

Since then, numerous studies have been published, which include a measure of kidney function as an outcome of interest [10–13,15,100–114]. The effect of exercise training on CKD progression (i.e., eGFR decline) has been further examined in several systematic reviews with meta-analysis [17,115–119], identified by a comprehensive literature search (see Supplementary Materials). These analyses are summarised in Table 1. As can be seen in Table 1, these reviews have reported conflicting findings, with almost equal numbers reporting a statistically significant, and clinically meaningful, beneficial effect on eGFR as no effect. This is despite, in the main, including the same studies (Supplementary Table S1). It should be noted that no published meta-analysis (or individual study) has shown a statistically significant reduction in kidney function in response to exercise training as compared to usual care, i.e., there is no detrimental effect. The results of the individual studies included in the systematic reviews, along with other relevant studies, are presented

in Supplementary Table S2. Here, we critically analyse the published systematic reviews to summarise the current state of play in this field of research, identifying the reasons for the disparate findings, in order to suggest future research directions.

One of the key reasons for the difference in results is the choice of which eGFR outcome is used in each meta-analysis. Some compare the eGFR at final follow-up between exercise and usual care groups; others compare the change in eGFR from baseline. Clearly, the analyses that only compare final eGFR do not take into account differences in eGFR at baseline; this is an issue for small studies where randomisation may not result in similar values between groups. This is certainly the case in Greenwood et al. [10]: the baseline eGFR (CKD-EPI$_{cr}$) was 36.6 in the exercise training group, compared to 46.5 in the controls. A similar discrepancy was seen in Headley et al. [13] (33.2 vs. 48.5); in other studies, the difference between exercise and control at baseline was 4 mL/min/1.73 m$^2$ or less. Whilst these differences should be addressed by the meta-analysis methodology, where individual study results are combined and potential discrepancies in the randomisation are addressed by the increased total sample size, it remains possible that the groups were not equal in eGFR at baseline. It is also worth noting that only one of the included studies [10], gives information about the rates of decline prior to the studies, which may also be different between treatment groups, as it was in that study; the pre-intervention rate of decline may affect the response to treatment.

Nearly all the reviews include multiple studies with overlapping participant samples. Both studies by Aoike et al. and those by Baria and Gomes et al. were all sub-populations of the same study; specifically, Aoike et al., 2015 [104] was a 12-week duration and only the home-based exercise training group; Aoike et al., 2018 [112] was a 24-week duration, Baria et al., 2014 [102] reported males only and Gomes et al., 2017 [108] was a complete study similar to Aoike et al., 2018 [112], but with the two exercise groups (home and centre) combined. Similarly, Howden et al., 2013 and 2015 [8,120] report the same sub-population of which Beetham et al., 2018 [114] had a larger sample. Thus, all the systematic reviews, except Nakamura et al., 2020 [116] were biased by including results from the same patients multiple times.

There are other differences in the studies included in these reviews, which may also contribute to the difference in meta-analysis results. In the main, this appears to be due to differences in study inclusion criteria (for example, [113] is not included as it only recruited hypertensive non-diabetics; Ref. [12] had a factorial design that included dietary intervention) rather than a failure in systematic review methodology or conduct. An exception is Wu et al. [117]. Despite the methodology specifying randomised and quasi-randomised controlled trials, they included two single-arm studies [105,106], four randomised controlled trials [10,107,109,113] plus three studies only available in Chinese [121–123]; notably, these three studies were not included in Nakamura et al., 2020 [116] or Zhang et al., 2019 [118] despite both these reviews including Chinese language articles and searching Chinese databases. Similarly, Zhang et al., 2019 [118] includes [98], which is not randomised.

**Table 1.** Summary of meta-analyses of effect of exercise on kidney function.

| Review Reference | Exercise Training | Inclusions | Studies (or Strata) (n) | Total Exercise Participants | eGFR Results Analysed | eGFR Difference (mL/min/1.73 m$^2$) |
|---|---|---|---|---|---|---|
| Yamamoto 2021 [115] | Aerobic (± resistance) | (e)GFR < 60 RCTs 1 month + duration English language Cardiometabolic or renal outcomes | 11 | 221 | Between-group mean final eGFR | 0.04 |
| Nakamura 2020 [116] | Aerobic or resistance or both | (e)GFR < 60 RCTs or cross-over 1 month + duration 1+ per week Intesity described No language restriction eGFR as outcome | 10 | 238 | Mixture of final eGFR and within-group difference in eGFR | −0.34 |
| Wu 2020 [117] | Combined aerobic and resistance | CKD 1–5 (non-dialysis) RCTs and quasi-RCTs Any combined exercise intervention Chinese or English language eGFR as outcome | 6 | 143 | Between-group mean final eGFR [a] | 5.01 * |
| Villanego 2020 [17] | Aerobic (± resistance) | CKD 1–5 (non-dialysis) assignment "described as random" 12 weeks + duration English or Spanish language | 13 | 226 | Within-group change, intervention minus control | −0.14 |
| Zhang 2019 [118] | Aerobic (± resistance) | CKD 2–5 (non-dialysis) RCTs 1 month + duration 1 + per week Chinese or English language eGFR as outcome | 18 | 262 | Within-group change, intervention minus control | 2.62 * |
| Vanden Wyngaert 2018 [119] | Aerobic (± resistance) | CKD 3–4 (non-dialysis) RCTs 3 month + duration 2 + per week Dutch, English, German or French language eGFR as outcome | 10 | 154 | Within-group change, intervention minus control | 2.16 * |

Findings of eGFR differences in meta-analyses of studies of exercise in ND-CKD. Positive numbers indicate a benefit to exercise training in reducing eGFR decline as compared with usual care. Statistically significant results are marked with an asterix (*). [a] Wu et al., 2020 also reported, in 9 studies with 329 participants, a statistically significant within-group benefit of exercise on eGFR of 3.01 mL/min/1.73 m$^2$.

Other relevant studies have been excluded from these reviews, including [15,96,99]. These all used measured GFR (by $^{51}$Cr-EDTA, $^{125}$I-iothalamate clearance and iohexol clearance, respectively) rather than eGFR, which was the outcome of interest in the systematic reviews. Castaneda et al., 2001 [99] may also have been excluded because it was a trial of resistance-only exercise training, i.e., no aerobic component; also, Ref. [15] compared two different forms of exercise training without a usual care control. In addition, a Chinese study [103] comparing Tai Chi to usual care in the non-dialysis CKD population was not included in any systematic review. This study showed a statistically significant benefit on eGFR (an increase of 6 mL/min/m$^2$ in the Tai Chi group). Presumably, this study was not included because Tai Chi is not widely considered as a form of exercise training, though it has been shown to have benefits on cardiorespiratory function, muscle strength and endothelial function [124,125].

Notwithstanding that the systematic reviews show contradictory results, there are other reasons to interpret their findings with caution. There is considerable heterogeneity in the study designs and the exercise interventions used, such that judgment about the generalisability of the findings is difficult. Additionally, the differences in study design reduce the legitimacy of combining the results by meta-analysis. Details of the included studies (and other studies with relevant outcomes) are found in Table 2. Firstly, there are differences in participant selection: whilst all selected patients with either CKD 2–4, 3 or 3–4, many studies specified other inclusion criteria, including requiring non-diabetic, type-2 diabetes mellitus (T2DM), hypertensive or obese patients or the presence of other cardiovascular risk factors. Secondly, there were important differences in the exercise interventions themselves. Some studies included light aerobic exercise only, including walking, for all or part of the programme [100,102,104,108,109,112]. Most others included a moderate aerobic component, but this was variably defined and prescribed: some studies used percentage of heart rate achieved during a maximal exercise tolerance test, others used the participant's own Rating of Perceived Exertion (RPE), a subjective measure of the intensity of the activity. One study used short bouts of high-intensity cycling training [11]. Some studies also incorporated non-exercise elements in the interventions, which were not offered to the control groups. These included dietary advice or prescription [8,13,107,114,120]. The duration of interventions ranged from 12 to 52 weeks, with just one lasting longer at 156 weeks [110]; notably, this study demonstrated significant improvements in eGFR in both exercise groups at the end of the intervention, though there was no non-exercise control to compare with.

As well as noting the marked variability in intervention duration, it is worth considering whether these time frames are sufficient to see a difference in eGFR between treatment groups. The typical rate of decline in people with established CKD can vary from as little 2.4 to 8.5 mL/min/1.72 m$^2$ per year [126]; the KDIGO international guideline defines >5 mL/min/1.72 m$^2$ per year reduction in eGFR as rapid decline. Furthermore, it is widely recognised that decline in eGFR is not linear; most patients with CKD demonstrate considerable variability in their disease course, often with a prolonged period of non-progression [1,127]. Furthermore, one might compare the length of follow-up required in landmark studies of pharmacological interventions. Follow-up durations of 2.4 to 3.4 years have been needed to reach composite primary end-points, which included doubling of serum-creatinine and progression to ESKD (markers of CKD progression), in studies of key agents, such as angiotensin-converting enzyme inhibitors (ACEi) angiotensin receptor blockers (ARB) and SGLT-2 inhibitors [128–133].

**Table 2.** Characteristics of exercise training studies in non-dialysis CKD, with renal function as an outcome.

| Study Reference | Groups (n) | Study Duration (Weeks) | Mean eGFR (Baseline) | Other Population Inclusion Criteria | Age (Years) | Exercise Frequency and Time | Aerobic Exercise | Resistance Exercise |
|---|---|---|---|---|---|---|---|---|
| Aoike 2015, 2018, Baria 2014, Gomes 2017 [102,104,108,112] | H-ET (14); C-ET (13); UC (15) | 12–24 | 26.7 | BMI > 25; male only in Baria 2014 | 55 | 30–50 min, 3 × per wk | Mild-Moderate (VT i.e., 40–60% $VO_2$max): home-based = walking; centre-based = treadmill | Nil |
| Barcellos 2018 [113] | ET (76); UC (74) | 16 | 69.0 | Hypertensive, non-diabetic | 65 | 60 min, 3 × per wk, supervised | Step aerobic workout in 30 s bouts as part of circuit training | Body weight, core and dumbbell exercises in 30 s bouts as part of circuit training |
| Beetham 2019 [7] | HIIT (9); Moderate ET (5) | 12 | 61.6 | 1+ uncontrolled CVD risk factors (BP, HbA1C, lipids) | 61 | HIIT: 4 × 4 min; Moderate Ex: 40 min 3 days per wk | HIIT: 80–95% PeakHR, Mod Ex: 65% peakHR; treadmill | Nil |
| Castaneda 2001/2004 [99,134] | ET + low-protein diet (14); low-protein diet only (12) | 12 | 29.5 | Age > 50 | 64 | 45 min, 3 × per wk | Nil | 3 sets × 8 reps, upper and lower body exercises using resistance training machines; 80% of 1RM |
| Eidemak 1997 [96] | ET (15); UC (15) | 78 | 25 | Non-diabetic | 44 | 30 min, daily; unsupervised | 60–75% maximal exercise capacity, static bike, running, swimming or walking | Nil |

**Table 2.** *Cont.*

| Study Reference | Groups (n) | Study Duration (Weeks) | Mean eGFR (Baseline) | Other Population Inclusion Criteria | Age (Years) | Exercise Frequency and Time | Aerobic Exercise | Resistance Exercise |
|---|---|---|---|---|---|---|---|---|
| Greenwood 2015 [10] | ET (8); UC (10) | 52 | 42.1 | | 53 | 40 , 3 × per wk (2 × supervised, 1 × home) | stationary exercise cycle; 80% HRR | 3 sets × 8–10 reps, upper and lower body exercises using free weights or resistance bands; 80% of 1RM |
| Gregory 2011; Headley 2012 [13,135] | ET (10); UC (11) | 48 | 41.2 | | 55 | 45 min, 3 × per wk, supervised | 50–60% $VO_2$peak | Weeks 24–48; 1–3 sets × 10–12 reps; upper and lower body using weight machines |
| Hamada 2016 [106] | ET (47) | 26 | 47.7 | | 69 | 90–120 min, 6 × per month | Walking, RPE 12–14 | 3–4 METs |
| Headley 2014, 2017, Miele 2017 [111,136,137] | ET (25); UC (21) | 16 | 47.6 | DM or HTN as primary cause of CKD | 58 | 30–45 min, 3 × per wk, supervised | 50–60% $VO_2$peak, mixed apparatus | Nil |
| Hellberg 2019, Zhou 2020 [15,138] | Endurance & strength (73); Endurance and balance (75) | 52 | 19.5 | | 66 | 150 min/ wk | 2 × 30 min; RPE 13–15 | 3 × 30 min; RPE 13–17; 2–3 sets × 10 reps; free weight/body weight (resistance) or balance exercises |
| Hiraki 2017 [109] | H-ET (14); UC (14) | 52 | 39.5 | Male | 68 | 30 min daily or 8000–10,000 steps, plus resistance | Walking | Hand grip, squats and calf raises; 20–30 reps 3× per week |

**Table 2.** *Cont.*

| Study Reference | Groups (n) | Study Duration (Weeks) | Mean eGFR (Baseline) | Other Population Inclusion Criteria | Age (Years) | Exercise Frequency and Time | Aerobic Exercise | Resistance Exercise |
|---|---|---|---|---|---|---|---|---|
| Howden 2013,2015, Beetham 2018, Small 2017, Huppertz 2020 [8,114,120,139,140] | ET and lifestyle intervention (81); UC (80) | 52 | 39.6 | 1 + uncontrolled CVD risk factors (BP, HbA1C, lipids) | 62 | 150 min per wk, beginning with 8 wks 2–3 × per wk supervised in gym; subsequently at home | Moderate, RPE 11–13 | Whole-body with therabands and Swiss ball |
| Ikizler 2018 [12] | ET vs usual activity; Calorie restriction vs. usual diet (104 total) | 16 | 42.4 | BMI > 25 | 57 | 30–45 min, 3 × per wk, supervised | 60–80% $VO_2$peak; cycling, treadmill or epliptical | Nil |
| Kirkman 2019 [141] | ET (15); UC (16) | 12 | 43 | | 58 | 45 min, 3 × per wk; supervised | 60–85% heart rate reserve; cycling, treadmill or epliptical | Nil |
| Kiuchi 2017 [110] | HIIT (25); Mod ET (25) | 156 | 43.1 | HTN | 58 | HIIT: 4–30 min; Mod Ex: 30–60 min 5 days per wk | HIIT: up to maximal intensity; Mod Ex: 55–85% max HR | Nil |
| Leehey 2016 [107] | ET + diet (14); Diet alone (18) | 52 | 39.9 | T2DM, obese, male | 66 | 60 min, 3 × per wk; superveised for first 12 wks | 45–85% $VO_2$peak; Treadmill, elliptical or cycle; intervals | Additional 20–30 min/ session |
| Leehey 2009 [100] | ET (7); UC (4) | 24 | 45 | Diabetic, BMI > 30, proteinuria | 66 | 30–40 min, 3 × per wk, supervised for first 6 wks | Walking; supervised sessions were up to 60–84% $VO_2$peak | Nil |

| Study Reference | Groups (n) | Study Duration (Weeks) | Mean eGFR (Baseline) | Other Population Inclusion Criteria | Age (Years) | Exercise Frequency and Time | Aerobic Exercise | Resistance Exercise |
|---|---|---|---|---|---|---|---|---|
| Mustata 2011 [101] | ET (10); UC (10) | 52 | 27.5 | | 68 | 20–60 min, 5 × per wk (3 unsupervised, 2 supervised) | 40–60% VO$_2$peak, treadmill, cycle, ellipitcal trainer; unsupervised = walking | Nil |
| Nylen 2015 [105] | ET (128) | 12 | 76.1 | T2DM | 62 | 60 min, 2 × per week | Mixed aerobic and resistance; 50–85% HRR | No details |
| Pechter 2003 [142] | ET (17); UC (9) | 12 | 65.3 | Proteinuria; 2 + CVD risk factors | 49 | 30 min, 2 × per wk | Aquatic exercise | Nil |
| Shi 2014 [103] | ET (11); UC (10) | 12 | 45 | Evidence of CVD | 69 | 30 min, 3–5 × per wk, supervised for first 4 wks | Tai Chi | Nil |
| Toyama 2010 [98] | ET (10); UC (9) | 12 | 47.4 | CVD | 72 | 30 min, 7 × per wk (1 × supervised, 6 × home) | 1 × supervised bicycle ergometer; daily walking, RPE 12–13 | Nil |
| Van Craenenbroeck 2015 [11] | ET (19); UC (21) | 12 | 38.6 | | 53 | 10 min, 4 × per day, at home | Static cycling, 90% HR of anaerobic threshold | Nil |

Abbreviations: BMI, body mass index; C-ET, centre-based exercise training; CVD, cardiovascular disease; DM, diabetes mellitus; ET, exercise training; H-ET, home-based exercise training; HIIT, high intensity interval training; HR, heart rate; HRR, heart rate reserve; HTN, hypertension; RPE, rating of perceived exertion; T2DM, Type 2 diabetes mellitus; UC, ususal care; VT, ventilatory threshold. Trials of exercise training in people with non-dialysis CKD, which include a measure of kidney function as an outcome. These studies are either included in recent meta-analyses or otherwised referenced in this review. Where multiple publications have been made from the same study (i.e., interim analyses or sub-samples), information for sample sizes, mean eGFR and age are taken from the most complete publication. All studies in the table were randomised-controlled trials except the following: Pechter 2003 and Toyama 2010 (non-randomised controlled trials); Hamada 2016 and Nylen 2015 (single-arm).

Mention should be made of the measures of kidney function used in these studies. As already alluded to, none of the included studies used directly measured GFR. The majority of studies included in the meta-analyses reviewed here use creatinine-based estimates of GFR with a variety of formulae (Supplementary Table S2). The influence of muscle mass and diet on serum creatinine concentrations [143] is of concern in exercise interventions where there is an expectation that muscle mass may increase. Cystatin C has been proposed as an alternative molecule that could be used for estimating GFR. Cystatin C is produced by all human nucleated cells and completely filtered at the glomerulus before being reabsorbed and metabolised by the tubular epithelium [144]. The use of cystatin C for estimating GFR is not without its own limitations due to some extra-renal elimination and influence of factors such as body mass, CRP and white blood cell count [145,146]. Cystatin C, however, appears to be less influenced by muscle mass [146,147], and so its use in exercise studies is preferable to creatinine-based estimates alone, although it can still lead to misinterpretation of kidney function in patients with very high or very low muscle mass [148]. Cystatin C-based measures were used in only two studies [12,114] (and as an exploratory analysis in [10]).

A key reason why studies may have failed to find a positive effect of exercise training, and another reason why it is difficult to draw conclusions on efficacy from the findings of the meta-analyses to date, is the issue of compliance. Compliance with an exercise programme, both in terms of attending exercise sessions and fidelity to the prescribed duration and intensity, is clearly important to assessing the efficacy of the training. These elements were another source of variability in the studies included in previous meta-analyses (if they were reported at all). The details of participant compliance are summarised in Table 3. Overall, where it is reported, compliance to session attendance was 70–97%, although it is likely to have been lower in some studies, including [12] (85% attended at least 50% of sessions) and [113] (63.8% attended at least 70% of sessions). Conversely, adherence/fidelity to the prescribed programme, i.e., to what extent—in terms of mode, intensity and time—did participants complete the prescribed exercise, is rarely reported. Study authors report using a variety of means to monitor the completion of prescribed exercise, including monitoring by completion of diaries. Heart rate monitors, which can give more accurate information about the duration and intensity of the exercise (when data is combined with that from a graded exercise tolerance test), were used in only a few studies and mainly without reporting of data (see Table 3). The baseline and follow-up testing of the $VO_2$peak acts as a manipulation check to assess if interventions have occurred and been adhered to sufficiently to induce physiological change. Six studies showed improvements in cardiorespiratory fitness [8,10,11,13,111,112]; thus, even though we do not know much about the fidelity of exercise programmes, it was sufficient for some physiological adaptation. In contrast, however, an equal number of studies did not include a measure of cardiorespiratory fitness [109,110,113] or reported no change [12,100,107] in this outcome measure.

This leads to an important question: even if all ET programmes were designed to induce improvements in cardiorespiratory fitness, and the programmes were sufficiently adhered to gain these benefits, is this the same level of intensity, frequency or duration of activity sufficient to induce physiological changes that benefit kidney function? Perhaps a more fundamental question is what do we expect exercise training to do, physiologically or immunologically or otherwise, that would delay kidney disease progression? This will be considered in the following sections.

**Table 3.** Measures of compliance in exercise training studies.

| Study Reference | Session Attendance Compliance | Exercise Intensity Compliance | Exercise Time Compliance | Effect on VO₂Peak (mL/kg/min) [*] |
|---|---|---|---|---|
| Aoike 2015, 2018, Baria 2014, Gomes 2017 [102,104,108,112] | 84–87% | *Not reported* | 78% | 2.3–5.4 |
| Barcellos 2018 [113] | *Not reported* [a] | *Not reported* | *Not reported* | *Not measured* [c] |
| Greenwood 2015 [10] | 79% | *Not reported* | *Not reported* | 5.7 |
| Headley 2012 [13] | 83% | 650 kcal/week (no target) | *Not reported* | 1.4 |
| Headley 2014, 2017, Miele 2017 [111,136,137] | 95–97% | *Not reported* | 78% | 1.6 |
| Hiraki 2017 [109] | *Not reported* | Target steps met | *Not reported* | *Not measured* |
| Howden 2013,2015 [8,120] | Supervised: 70% | *Not reported* | 57% | 2.8 |
| Ikizler 2018 [12] | *Not reported* [b] | *Not reported* | *Not reported* | No significant difference |
| Kirkman 2019 [141] | 92% | "Completed as prescribed" | "Completed as prescribed" | 2.09 |
| Kiuchi 2017 [110] | *Not reported* | *Not reported* | *Not reported* | *Not measured* |
| Leehey 2016 [107] | *Not reported* | *Not reported* | *Not reported* | No significant difference |
| Leehey 2009 [100] | *Not reported* | *Not reported* | *Not reported* | No significant difference |
| Van Craenenbroeck 2015 [11] | 95% | 101% of HR target | *Not reported* | 5.82 |

Measures of compliance in exercise training studies included in meta-analyses of effects on kidney function. Intensity compliance refers to measures such as energy expenditure and heart rate (HR). Time refers to time per exercise session; [a] "70% reached in 63.8%"; [b] "85% completed at least 50%"; [c] Improved sit-to-stand 30 s; [*] Numeric results indicate statistically significant findings in the exercise training group.

## 5. CKD Risk Factors Are Amenable to Exercise Interventions

There are a number of possible ways in which exercise may help delay CKD progression. Importantly, the CKD risk factors discussed above have been shown to be amenable to exercise interventions. The benefit of exercise training has been demonstrated in other disease populations, including improved blood pressure in elderly hypertensives and other groups [149,150]; improved glycaemic control in T2DM [151]; improved insulin sensitivity in T2DM and other populations [152,153]; and weight loss in overweight and obese adults [154].

There is also evidence that some of these risk factors can be successfully targeted with exercise training in patients with CKD. Evidence includes a number of systematic reviews with meta-analyses demonstrating positive effects on blood pressure and BMI [17,115,118,119,155,156], summarised in Tables 4 and 5. Though not all statistically significant, a clinically meaningful mean reduction in systolic blood pressure (SBP) has been found in all these meta-analyses. The exception is Van den Wyngaert et al., 2018, though that review also noted a significant within-group reduction in SBP of 5.2 mmHg following exercise training. Again, the reviews analysing the effects on BMI find mean differences in favour of exercise training; in the main, these results are statistically significant. The effects of exercise training on other factors in CKD patients have been less well studied. The effect on glycaemic control has been reviewed, and no effect was found on HbA1C in one meta-analysis, which included seven studies and sub-strata [115]; mixed results were found in another systematic review [157].

**Table 4.** Effect of Exercise on Blood Pressure in ND-CKD: summary of meta-analyses results.

| Review Reference | SBP Analysis | Result (mmHg), ET vs. UC |
|---|---|---|
| Zhang 2019 [118] | Final follow-up | −5.61 (−8.99, −2.23) |
| Thompson 2019 [155] | Final follow-up | −4.33 (−9.04, 0.38) |
| Villanego 2020 [17] | Baseline to follow-up change | −1.68 (−6.80, 3.44) |
| Yamamoto 2021 [115] | Final follow-up | −0.75 (−1.24, −0.26) |
| Van den Wyngaert 2018 [119] | Baseline to follow-up change | 1.22 (−4.45, 6.90) |

Effects of exercise training on systolic blood pressure (SBP) as found in meta-analyses of studies in patients with non-dialysis CKD (ND-CKD). Results are the mean difference (plus 95% CI) between exercise training (ET) and ususal care (UC) groups, either the SBP result at final follow-up, or the change from baseline to final follow-up.

**Table 5.** Effect of Exercise on body mass index in ND-CKD: summary of meta-analyses results.

| Review Reference | BMI Analysis | Result (kg/m$^2$), ET vs. UC |
|---|---|---|
| Zhang 2019 [118] | Final follow-up | −1.32 (−2.39, −0.25) |
| Villanego 2020 [17] | Baseline to follow-up change | −0.89 (−1.47, −0.31) |
| Wu 2021 [156] | Final follow-up | −0.77 (−1.31, 0.23) |
| Yamamoto 2021 [115] | Final follow-up | −0.19 (−0.38, 0.0) |

Effects of exercise training on body mass index (BMI) as found in meta-analyses of studies in patients with non-dialysis CKD (ND-CKD). Results are the mean difference (plus 95% CI) between exercise training (ET) and ususal care (UC) groups, either the BMI result at final follow-up, or the change from baseline to final follow-up.

Whilst meta-analyses have found positive effects on both BMI and BP in exercise studies in non-dialysis CKD, these findings do not consistently correlate with improved eGFR outcomes: e.g., Ref. [119] found a positive effect on eGFR, with no effect on BP, whilst [115] found positive effects on BMI and BP but not eGFR. There is a similar mismatch in the individual studies reviewed in these meta-analyses.

## 6. Potential Benefits of Exercise on the Pathological Processes of CKD Progression

We have seen above that exercise training in CKD patients produces physiological adaptations that produce improved cardiorespiratory fitness and muscle strength. It is reasonable to consider whether similar training may induce adaptations in the kidney.

Whilst the precise mechanisms have not been elucidated, the stimulants for exercise-induced physiological adaptations occur primarily due to muscle stretching. Muscle activity results in changes including intra-muscular energy consumption, intracellular pH and $Ca^{2+}$ concentration, as well as the generation of mechanical forces by muscle contraction [158]. These stimulants have all been linked to changes in protein and enzyme transcription, which result in metabolic, hormonal and vascular changes, which then produce beneficial adaptations to exercise training. There are also stimulants that occur outside of the contracting muscles, including increased shear stress driven by increased cardiac output and blood flow and altered sympathetic nervous activity [159]. Clearly, physical stretching is not relevant to kidney tissue. Furthermore, during exercise, renal blood flow diminishes rather than increases. Hence, mechanisms specific to muscles (related to mechanical stretching and increased shear stress) are not replicated in the kidney. However, whilst these stimulants derive from active muscle tissue, the downstream effectors are not necessarily limited to muscles, and some are of relevance to renal pathophysiology. For example, skeletal muscle capillary growth is stimulated by mechanical influences, such as shear stress induced by increased blood flow, and stretching of the muscle tissue, with VEGF mediating the effect [160,161]. VEGF may have systemic and beneficial effects on maintaining the renal vasculature, particularly the glomerular and tubular capillaries, which are so relevant to the pathogenesis of CKD progression. There are a number of lines of evidence to suggest that the physiological changes that occur during exercise not only occur in the active skeletal muscle but also take place elsewhere in the body. Examples include an improvement in endothelial-dependent dilation in the brachial artery as a result of leg cycling training [162]. Distal vascular changes have also been demonstrated in studies where there is no increased

cardiac load: in a four-week passive leg movement programme, there was an increased capillary density and VEGF concentration in the muscle of the untrained leg [163].

Another key adaptation to exercise training is mitochondrial biogenesis [164]. This is an important development, in response to $Ca^{2+}$ cycling and ROS production etc., for improving the energy supply to active muscle. It would also be a useful adaptation in metabolically active kidney tissue, helping to optimise the use of oxygen to produce energy, potentially alleviating the effects of hypoxia in CKD pathophysiology. In animal models of diabetic and hypertensive nephropathy, aerobic exercise training produced beneficial effects on renal mitochondria, including inducing key enzymes and transcription factors for mitochondrial biogenesis, increasing mitochondrial ATP and reducing mitochondrial ROS production. These changes were associated with reduced renal disease progression compared with untrained animals [165–167].

Studies in humans have demonstrated reduced concentrations of some of the key molecules implicated in renal pathophysiology (discussed above) in response to exercise. These include MCP-1 [168,169], RANTES [170], NF-κB [171] and TNFα [172]. Exercise also increases anti-oxidants and decreases pro-oxidants in a variety of healthy and chronic disease populations [173]. Alterations to levels of myokines, i.e., cytokines produced by active skeletal muscle tissue, may also be relevant because of their anti-inflammatory and metabolic effects. These include the effects of IL-6 on increasing fatty acid oxidation, thus increasing energy availability, and reducing TNFα production from macrophages, amongst other anti-inflammatory effects [174]. Another example is the myokine irisin, serum levels of which are decreased in patients with CKD [175,176] and can be increased with exercise training, e.g., in older adults [177]. Interestingly, a mouse model of CKD (interstitial fibrosis) demonstrated improved kidney mitochondrial energy metabolism and reduced fibrosis after irisin induction [178]. Whilst some of these changes have not yet been demonstrated in patients with CKD, exercise may help readjust the imbalance of these important mediators in this population.

Benefits on other inflammatory and oxidative stress molecules have been seen in patients with CKD: 12 weeks of exercise in CKD 3–4 ameliorated cutaneous microvascular endothelial dysfunction by reducing oxidative stress. This suggests systemic anti-oxidant and endothelial function benefits of exercise [141]. Similarly, exercise training in CKD has been shown to reduce serum markers of oxidative stress, including $F_2$-isoprostane [12] and reduce lipid peroxidation and glutathione oxidation after 12 weeks of aquatic exercise in participants with mild-to-moderate CKD [142]. There is reduced (sodium oxide dismutase) SOD and Nrf2, important anti-oxidant/anti-inflammatory molecules in moderate CKD, compared with healthy controls; this was improved after acute exercise [179]. Likewise, resistance exercise in haemodialysis patients increases Nrf2 [180]. These are systemic anti-oxidant effects that may benefit ROS imbalance present in the diseased kidney.

It is important to consider the extent to which improved systemic concentrations of circulating molecules are reflective of intra-renal levels: i.e., does a systemic reduction in, for example, anti-oxidants after exercise translate into an improvement in oxidative stress in the glomerulus or tubules of the kidney? Such questions would require further animal studies to develop our understanding in this area.

Finally, whilst a bout of exercise leads to increased SNS activity [181], exercise training is known to dampen this response and may even reduce resting sympathetic tone [182]. Indeed, exercise training in healthy adults has been shown to result in reduced resting renal sympathetic activity, specifically [183]. As discussed, CKD patients have SNS overactivity, which may contribute to the pathophysiology of the disease. Thus, reduced SNS activity may be another way in which exercise training may affect CKD progression.

In summary, there are a number of potential ways in which exercise may reduce CKD progression. Prominent risk factors for CKD—BP and BMI—are improved by exercise, both in CKD patients and other populations. It is also plausible that exercise may affect the mediators of kidney disease pathophysiology themselves, with human exercise studies showing modification of many of the relevant molecules and pathways.

## 7. Summary and Discussion of Future Directions

Overall, the story so far is that, despite benefits on disease progression risk factors, the effect of exercise on kidney function itself is unclear. The heterogeneous findings of recent meta-analyses reflect the heterogeneity of the method of analysis and the studies included. Alongside this, in preventing a definitive conclusion, is the heterogeneity of individual study designs, particularly in terms of duration and intensity of exercise interventions. Uncertainty about participants' compliance to the exercise programme also makes assessment of the efficacy difficult.

We have seen that it is probably unreasonable, given the typical rates and patterns of eGFR decline, to expect to see significant or meaningful differences after durations of exercise training less than 12 months. Recently, consideration has been given to using the slope of eGFR decline as an outcome in clinical trials [184–186] with a suggestion that slope over 2–3 years is a useful surrogate marker for more definitive end-points, such as reaching ESKD. We might also consider whether it is realistic to expect to meaningfully delay the rate of kidney function decline in patients with already advanced disease. More advanced CKD stage is consistently shown to be a risk factor for more rapid eGFR decline [60,84,187,188]. This is likely to be one reason why pharmacological intervention studies, such as those discussed herein recruit patients with a wider range of eGFR and often exclude those with more advanced CKD. Those designing future studies in this field need to consider who is likely to benefit: if patients are already at an advanced stage with progressive disease, is it too late to intervene? Intervention in earlier stages of CKD could be considered, but this would increase the chance of intervening on patients at low risk of decline who would not progress anyway. An alternative may be to select those at earlier stages of CKD with risk factors for kidney function decline.

As discussed, neither creatinine or cystatin C provide perfect estimates of GFR. Hence, consideration might be given to using directly measured GFR. Measuring GFR using exogenous filtration markers—such as $^{51}$Cr-EDTA, $^{125}$I-iothalamate clearance and iohexol clearance—has a strong rationale for studies specifically investigating effects on kidney function. These measures, however, are not without their own limitations, including a degree of intra-test and inter-test variability in measurements, more prominent in people with higher GFR [143]. Perhaps more critical is the practicality and patient acceptability of these tests, which may only be available in specialist centres and, in most cases, take a number of hours and multiple venepunctures to perform. In our experience, there has been some patient reluctance because of the travel and time involved. A recently updated GFR estimating formula, using both serum creatinine and cystatin C measurements, has shown good accuracy with a median difference between measured GFR and eGFR of 0.1 and $-2.9$ mL/min/1.73 m$^2$ in black and non-black participants, respectively, and 91% of all estimates being within 30% of measured GFR [189]; accuracy was greater with lower GFR. In our opinion, eGFR using cystatin C in combination with creatinine is a practical and sufficiently accurate alternative to measured GFR for use in exercise studies, particularly in patients with eGFR < 60 and particularly if repeated measures of eGFR are available to calculate a slope of decline.

A large proportion of the studies included in the meta-analyses discussed here are relatively small and not designed to have power to detect differences in eGFR and are at risk of unequal groups at baseline. Therefore, as well as considering increased length of intervention, studies with larger sample sizes are needed to allow more robust randomisation; this is an important priority for future research. Multi-centre collaboration should be considered as a means to increasing sample size in studies whose primary outcome is change in kidney function.

Multiple barriers to exercise participation exist in patients living with CKD [190,191], so compliance is likely to always be an issue. This complicates assessing the efficacy of prescribed programmes, though this may reflect the effectiveness of exercise training in the real-world. Nevertheless, thought needs to be given to improving motivation to and accessibility of exercise training programmes to aid increased compliance, ideally without

compromising on the physiological load so that studies continue to address the underlying scientific question. The mechanisms involved in exercise effects on kidney function remain to be seen, and this, too, deserves further attention. Our opinion is that the highest priority for future research into the effects of exercise on CKD progression should be a more targeted approach to patient selection, i.e., identifying those patients who are most likely to benefit. This entails (1) identifying patients who are at risk of disease progression, and (2) targeting populations in which the adaptations induced by exercise training are likely to be most impactful; this requires a better understanding of the mechanisms of kidney adaptation to exercise.

Current recommendations are that patients with ND-CKD should complete 150 min of moderate aerobic exercise per week plus resistance and flexibility exercise on at least 2 days per week [192]. Whilst it is not clear which, if any, of these forms of exercise are more likely to lead to improvements in kidney function decline, we believe future studies in this field should incorporate both aerobic and resistance exercise into their programmes for several reasons. Firstly, studies that have shown benefits on CKD risk factors have invariably included aerobic, and often resistance, exercise. Secondly, some of the mechanisms described above for altering inflammatory levels are dependent on muscle stretch or have been demonstrated in resistance-only programmes. Thus, resistance training should also be included. Finally, both of these forms of exercise have demonstrated benefits to CKD unrelated to kidney function and it might be considered inappropriate to omit one or the other.

There is uncertainty about the benefits of exercise on disease progression in CKD. Whilst no RCT to date has shown a detrimental effect, it would be inappropriate at this stage to commit to exercise training for this purpose at the level of service commissioning or national guideline recommendations. Despite this, as there is no evidence of harm from exercise intervention on kidney function, and considering the many other potential benefits of increased physical activity, exercise should be increasingly recommended and prescribed by health professionals. Future research, including cost effectiveness analyses may then provide sufficient evidence to support exercise training being delivered as part of routine care.

**Supplementary Materials:** The following supporting information can be downloaded at: https://www.mdpi.com/article/10.3390/kidneydial2020026/s1, Table S1: Studies included in published systematic reviews of effects of exercise on kidney function in ND-CKD. Table S2: Effect of Exercise Training on Kidney Function in NDD-CKD: summary of study results.

**Author Contributions:** Conceptualization, M.D., J.M. and A.S.; writing—original draft preparation, M.D.; writing—review and editing, M.D., J.M. and A.S. All authors have read and agreed to the published version of the manuscript.

**Funding:** Mark Davies's PhD research is supported by Awyr Las, Registered Charity No. 1138976.

**Institutional Review Board Statement:** Not applicable.

**Informed Consent Statement:** Not applicable.

**Data Availability Statement:** Not applicable.

**Conflicts of Interest:** The authors declare no conflict of interest.

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
