# Peer review of "The Role of Exercise Training in Delaying Kidney Function Decline in Non-Dialysis-Dependent Chronic Kidney Disease"

_kidneydial, doi:10.3390/kidneydial2020026_

Round 1

Reviewer 1 Report

This is a comprehensive and well-done state of the science review on the effect of exercise on the progression of CKD from a mechanistic and clinical perspective. I commend the authors and I enjoyed reading this paper. I have several comments that I think would strengthen aspects of the review. 

  1. This is a narrative review and would benefit from more insight into what the authors think the areas of study are most important and the highest area for study at this time. For example, are studies with larger sample sizes needed right now or do we need more mechanistic work to identify highest risk groups most likely to respond i.e. precision-based approaches? In any case, it would be helpful for them to prioritize their recommendations. 
  2. There is a discussion of kidney function measurement (starting line 411). However, as alluded to earlier in the paper, few contemporary studies have directly measured GFR. Could the authors recommend or comment on this issue? Similarly, what are the issue with using muscle based measures of kidney function in exercise  training studies? Have any studies used Cystatin C?
  3. Suggest that line 344 is not a new paragraph. The next point follows from the previous idea that effect may be primarily downstream and the transition to a new paragraph read awkwardly. 
  4. In Section 6, I though the authors could elaborate on other potential mechanisms i.e. myokines with respect to inflammatory modulation especially given this is highlighted is sections 2&3.
  5. Suggest briefly mentioning the limitations to observational studies cited in lines 36 as 37 (selection bias and residual confounding) with respect to the effect of exercise on survival. 

Author Response

The authors would like to thank the reviewers for taking time to review our work, and for their comments and suggestions.

All the reviewers comments have been copied below and have been responded to in turn. Our responses are made in blue italics and a line number is provided for the relevant section in the updated manuscript.

1. This is a narrative review and would benefit from more insight into what the authors think the areas of study are most important and the highest area for study at this time. For example, are studies with larger sample sizes needed right now or do we need more mechanistic work to identify highest risk groups most likely to respond i.e. precision-based approaches? In any case, it would be helpful for them to prioritize their recommendations. 

Response 1. An opinion has been added on the prioritisation of targeting patient who are likely to benefit (line 489ff)

2. There is a discussion of kidney function measurement (starting line 411). However, as alluded to earlier in the paper, few contemporary studies have directly measured GFR. Could the authors recommend or comment on this issue? Similarly, what are the issue with using muscle based measures of kidney function in exercise  training studies? Have any studies used Cystatin C?

Response 2. A comment has been added, in the critique of studies (line 281), on the predominant use of creatinine-based eGFR (details given in Supplementary table 2), the preferable use of cystatin C. Additionally an opinion on the use of eGFR vs measured GFR is given in the summary (line 458ff)

3. Suggest that line 344 is not a new paragraph. The next point follows from the previous idea that effect may be primarily downstream and the transition to a new paragraph read awkwardly. 

Response 3. Changed as suggested

4. In Section 6, I though the authors could elaborate on other potential mechanisms i.e. myokines with respect to inflammatory modulation especially given this is highlighted is sections 2&3.

Response 4. Additional brief mention of myokines is added, specifically IL-6 and Irisin and their potential relevance (line 395ff)

5. Suggest briefly mentioning the limitations to observational studies cited in lines 36 as 37 (selection bias and residual confounding) with respect to the effect of exercise on survival. 

Response 5. As suggested a comment has been made on these potential limitations, lines 37 and 38.

Reviewer 2 Report

The study of Davies et al is narrative review focusing on currrent evidences about the effectiveness (if any) of exercise training in preserving kidney function  in patients with chronic renal failure. Overall the papers is very interesting and, to my opinion, well written. Current limitations of knowledge are properly underlined

I only have few suggestions for the authors

Current evidences are well reported however the database search should be better specified

Paragraph 3. Risk factors for CKD progression. This is an important paragraph since the effect of exercise training are mediated, at least in part, by the effects on CKD risk factors. In this context I suggest a short mention to blood pressure variability, since it has recently been associated to CKD progression (see Wang JAHA, 2020) – (’m not one of the auhors).

To my opinion among different variables affecting the results of reported trials, we should also considerer the volume of exercise performed by patients. Therefore I suggest that time, intensity number of repetitions (for resistance exercise) be reported either in the text or (better) in tables

Author Response

The authors would like to thank the reviewers for taking time to review our work, and for their comments and suggestions.

All the reviewer's comments have been copied below and have been responded to in turn. Our responses are made in blue italics and a line number is provided for the relevant section in the updated manuscript.

1. Current evidences are well reported however the database search should be better specified
Response 1. Details of the search strategy have been added to the supplemental material

2. Paragraph 3. Risk factors for CKD progression. This is an important paragraph since the effect of exercise training are mediated, at least in part, by the effects on CKD risk factors. In this context I suggest a short mention to blood pressure variability, since it has recently been associated to CKD progression (see Wang JAHA, 2020) – (’m not one of the auhors).

Response 2. Added as suggested (see lines 115ff)

3. To my opinion among different variables affecting the results of reported trials, we should also considerer the volume of exercise performed by patients. Therefore I suggest that time, intensity number of repetitions (for resistance exercise) be reported either in the text or (better) in tables

Response 3.  Details of exercise interventions are included in table 2. Following this comment, additional details of prescribed resistance exercises, where provided in the papers, have been added to that table.